METHODS

# A toolkit for the dynamic study of air sacs in siamang and other elastic circular structures

**Lara S. Burchardt**[1,2]*, **Yana van de Sande**[1], **Mounia Kehy**[3], **Marco Gamba**[4], **Andrea Ravignani**[5,6,7], **Wim Pouw**[1]*

**1** Donders Institute for Brain, Cognition, and Behaviour, Radboud University, Nijmegen, Netherlands, **2** Leibniz-Zentrum Allgemeine Sprachwissenschaft, Berlin, Germany, **3** Equipe de Neuro-Ethologie Sensorielle, Université Jean Monnet, France, **4** Department of Life Sciences and Systems Biology, University of Turin, Turin, Italy, **5** Comparative Bioacoustics Group, Max Planck Institute for Psycholinguistics, Nijmegen, Netherlands, **6** Center for Music in the Brain, Department of Clinical Medicine, Aarhus University & The Royal Academy of Music, Aarhus, Denmark, **7** Department of Human Neurosciences, Sapienza University of Rome, Rome, Italy

\* l.s.burchardt@gmx.de (LSB); wim.pouw@donders.ru.nl (WP)

**Data Availability Statement:** The computational tools can be found in our Github repository (https://github.com/WimPouw/AirSacTracker/tree/main). The code and data for reproducing the

## Abstract

Biological structures are defined by rigid elements, such as bones, and elastic elements, like muscles and membranes. Computer vision advances have enabled automatic tracking of moving animal skeletal poses. Such developments provide insights into complex time-varying dynamics of biological motion. Conversely, the elastic soft-tissues of organisms, like the nose of elephant seals, or the buccal sac of frogs, are poorly studied and no computer vision methods have been proposed. This leaves major gaps in different areas of biology. In primatology, most critically, the function of air sacs is widely debated; many open questions on the role of air sacs in the evolution of animal communication, including human speech, remain unanswered. To support the dynamic study of soft-tissue structures, we present a toolkit for the automated tracking of semi-circular elastic structures in biological video data. The toolkit contains unsupervised computer vision tools (using Hough transform) and super-vised deep learning (by adapting DeepLabCut) methodology to track inflation of laryngeal air sacs or other biological spherical objects (e.g., gular cavities). Confirming the value of elastic kinematic analysis, we show that air sac inflation correlates with acoustic markers that likely inform about body size. Finally, we present a pre-processed audiovisual-kinematic dataset of 7+ hours of closeup audiovisual recordings of siamang (*Symphalangus syndacty-lus*) singing. This toolkit (https://github.com/WimPouw/AirSacTracker) aims to revitalize the study of non-skeletal morphological structures across multiple species.

## Author summary

Many animals move not only by activating skeletal muscles, but also by pushing air or fluids into and out of cavities like air sacs. Such cavities can become part of the visible signals emitted by a species and may perform several functions, such as for example sound production or gas exchange. A most notable example of elastic cavities that move in the context of communication is the laryngeal air sac in a small Asian ape called the siamang.

kinematic acoustic analyses can also be found on our github page (https://github.com/WimPouw/AirSacTracker/tree/main/Project). The open dataset can be found on the Donders repository (https://doi.org/10.34973/6apg-q804).

**Funding:** The Comparative Bioacoustics Group is supported by Max Planck Independent Research Group Leader funding to AR. The Center for Music in the Brain is funded by the Danish National Research Foundation (DNRF117) awarded to AR. AR is funded by the European Union (ERC, TOHR, 101041885). WP is funded by a VENI grant (VI. Veni 0.201G.047: PI WP), Language in Interaction, and was further supported by the Donders Postdoctoral Development fund. The funders had no role in study design, data collection and analysis, decision to publish, or preparation of the manuscript.

**Competing interests:** The authors have declared that no competing interests exist.

There are currently a lot of open questions about the function of these air sacs for primates in general, but the exceptionally visible air sacs in siamang have never been dynamically studied to gain a better understanding of air sacs and their role in vocalizations and communication. To date, no adapted technology exists to track dilating movements of these inflating balloon-like structures. Here we provide a method to perform tracking of elastic circular structures alongside an open access data set of singing siamang with close-ups of the air sac. The techniques overviewed here can be applied to other inflatable structures that are abundant in other animals too.

## Introduction

Animal *skeletal* pose tracking from video data has undergone nothing short of a revolution through developments in computer vision and deep learning [1–6]. There are now many pre-trained pose detection models for humans, and there are increasingly more pre-trained models for non-human animals (e.g., rhesus macaques, *Macaca mulatta*, [7], chimpanzees, *Pan troglodytes* [8,9]). These developments are key for animal research as they allow non-invasive monitoring based on video alone, which can further be used for automatic classification of behavioral patterns [10–12]. The tracking of *elastic* biological structures has received comparatively less attention, leaving out a major aspect of biological motion. Organisms and other biological morphologies combine elements that resist compression (e.g., bones, skeleton, cytoskeleton) and elements that resist expansion (e.g., muscles, connective tissues, membranes) which include elastic structures such as shown in Fig 1 [13]. Together these elements form coherent interconnected systems that define how animals move and communicate

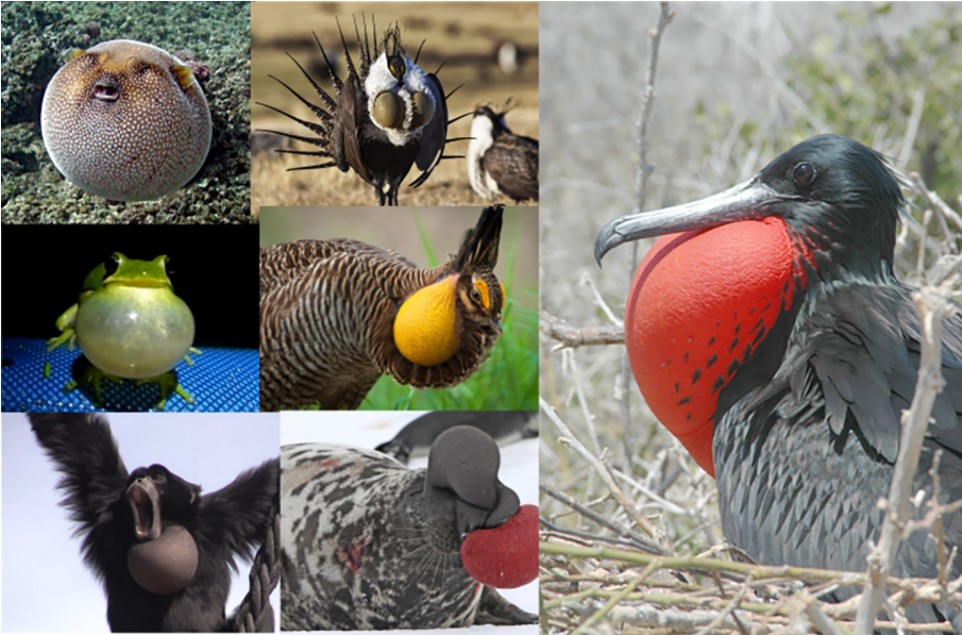

**Fig 1. Overview of elastic semi-circular biological structures.** Examples of different species with elastic endogenous and exogenous biological structures that are trackable as semi circles with the approaches described in this report. From top-left to top right, guineafowl puffer (*Arothron meleagris*), greater sage-grouse (*Centrocercus urophasianus*), green tree frog (*Hyla cinerea*), prairie chicken (*Tympanuchus cupido*), magnificent frigatebird (*Fregata magnificens*), siamang (*Symphalangus syndactylys*), elephant seal (*Mirounga angustirostris*). All photos are public domain.

[13,14]. These elastic structures include vocal sacs in frogs, tissue inflation in puffer fish, the gular sacs of birds, or air sacs in primates, such as the very prominent laryngeal air sacs in the siamang (*Symphalangus syndactylus*).

Elastic structures are involved in a wide variety of behaviors and ecological functions. In many species the expandable, oftentimes semi-circular structures are involved in communication, both acoustically and visually [15,16]. Such expandable structure may serve a function in mate attraction [16–19]. In birds they might also be involved in thermoregulation and stem from respiratory adaptations [20]. Monitor lizards and other reptiles have an inflatable gular cavity that aids in respiration [21] and may serve social functions (e.g., mate attraction). Many of these suggested functions remain empirically untested. There are also no detailed studies of dynamic shape and size variation of these inflatable biological structures. Here we show how focusing on elastic structures provides a first step towards a new layer of morphometry analyses, thus far mostly focused on skeletal, bony structures, which in turn can bridge knowledge gaps.

In the particular case of laryngeal air sacs, there is a lack of tools and data that are needed for empirical research in this domain to thrive. This is not for a lack of interest and there seems to be a gap of methodological and theoretical knowledge when it comes to air sacs. For example, understanding the function and functioning of air sacs is widely accepted as relevant to understanding the evolution of animal communication, including human speech [15,17,22–24]. In this paper we bridge this gap and provide a toolkit for tracking elastic kinematics via I) a data archive for audiovisual recordings of siamang air sac during their singing, II) a computer vision pipeline for analyzing spherical biological structures and air sacs, and III) proof of concept analyses linking air sac inflation with vocal acoustics. Our toolkit and proposed pipeline can enable the study of elastic biological structures in many more systems. They can provide indirect mechanistic insights via purely behavioral, non-invasive approaches.

## Small asian apes, the siamang, and laryngeal air sacs

Small asian apes or gibbons (*Hylobatidae*) are phylogenetically closely related to humans and other great apes [25]. Like humans, they are highly vocal [26–30]. Gibbons produce daily duetting songs to maintain and advertise pair bonds within an area, regulating their socially monogamous and territorial lifestyles [31]. These vocalizations are produced at high intensities and show highly distinctive species specific traits [26]. The siamang, also a gibbon species, can sing louder than 120 dB [32], thereby exceeding the vocalization ranges of most humans in terms of amplitude. Important for our current purposes, the siamang has one of the largest air sacs in extant primates relative to body size [24,25,33], and it is also the only species with a very visible semi-circular air sac. The air sac consists of a soft-walled cavity connected to the vocal tract just above the vocal folds and below the false vocal folds [33]. The soft-wall cavity forms a membrane under tension and thereby can resonate and radiate sounds. The air sacs are inflated during, and possibly in preparation for certain types of calls during singing, suggesting their possible supportive role in vocal production [33,34]. Some suggest [22,35,36] that air sacs inflate due to lung exhalation while closing the nasal and nostril passages, others hypothesize that the false vocal folds may close completely to redirect exhaled air into the air sac even when the upper air passages are open ([22], p. 635 for a discussion).

Primate laryngeal air sacs likely evolved because of some adaptive function. This hypothesis is tentatively supported by a cost-benefit analysis: why would air sacs evolve without a function when they also entail the risk of infections? Infections have been reported in detail in a mountain gorilla [37], and observed more generally in primates [38,39]. There are however no systematic studies that quantitatively report how often air sacs get infected. On the benefit side, there are many hypotheses about the function of (siamang) air sacs [17,23,33,34,40,41]; they

may: be relatively functionless [42], help manage oxygenation [41], or help stabilize the thorax during brachiation and singing [40]. Moreover, it has been suggested that an inflated air sac increase the expiratory flow relative to expiring from a single air reservoir (the lungs) when the upper air passages open, leading to a "glottal-shock" that increases the amplitude of the vocalization after the air sac is loaded with air [33].

Limited empirical work and real-world data are available to evaluate these hypotheses. Based on biomechanical and acoustic modeling we would expect a higher air sac volume to statistically predict less energy at higher formant frequencies relative to the dominant frequency and increased overall amplitude [34]. The latter relates to dynamic anti-resonance properties of the air sac (a cancellation of energy at resonant frequencies of the air sac), as supported by physical models and simulations that assess different static sizes of air sacs [22,34]. Thus the laryngeal air sac may likely serve as an amplifying organ by changing the resonant properties of the vocal source, whereby energy at lower frequencies is increased relative to the harmonics, thereby aiding sound travel and "dishonestly" advertising a larger body size [18,23]. In cluttered environments where formant structure can be lost even at short distances, body size might rather be signaled through longer sound duration, which could be enabled by the extra air volume that the air sacs provide (next to the lungs).

In sum, laryngeal air sac mechanics, articulatory states, and vocalization acoustics, feature many key unknowns. Studying the dynamic variation of air sacs together with articulation and acoustics will provide novel insights of their possible adaptive functions [43]. It promises to better understand the development of singing in the siamang by allowing to track air sac use and growth, and inter-individual variability therein. Further, by accounting for vocal variation attributed to air sac dynamics, we can start to better account for variations in vocal acoustics across species [24,44,45]. Indeed it was suggested that understanding the adaptive functions of air sacs is key to understanding the evolution of vocal-articulatory communication in hominins [46]. It is currently unknown why the laryngeal air sacs seem to have been lost in the genus Homo [47,48]. The evolutionary vestiges of air sacs are still present in humans, as is evident in pathological cases of trumpet players that develop a highly similar laryngeal air cavity [49]. Fitch concludes on these open issues that "Understanding why we lost air sacs requires a clear understanding of their function" [46] (p. 266). To gain such understanding we believe I) computer vision methods need to be optimized for elastic tracking, II) audiovisual data of laryngeal air sacs need to become available, and III) multimodal signal processing methods need to then be applied to study the relation of vocalization acoustics, articulatory and air sac kinematics. In this article, we provide a complete toolkit that fulfills all these requirements.

## Summary of our toolkit

The current toolkit includes a data archive, computer vision tools, and bioacoustic analysis. We first introduce I) an open dataset of more than seven hours, featuring tracking data, that allows for the detailed study of siamang air sacs. We then introduce II) a set of computer vision and data wrangling tools to track siamang air sacs and other spherical biological structures. Having introduced the toolkit, we report on promising findings that relate siamang air sac inflation with the acoustic properties of singing (III). The current paper provides a complete resource to promote a more in-depth study of the laryngeal air sacs and their functions (Fig 2).

## Materials and methods

### Ethics statement

The study was assessed and approved by the ethical committee of the Donders Institute (reference number: DCC.2022.00071), and the study was waivered for further ethical approval for

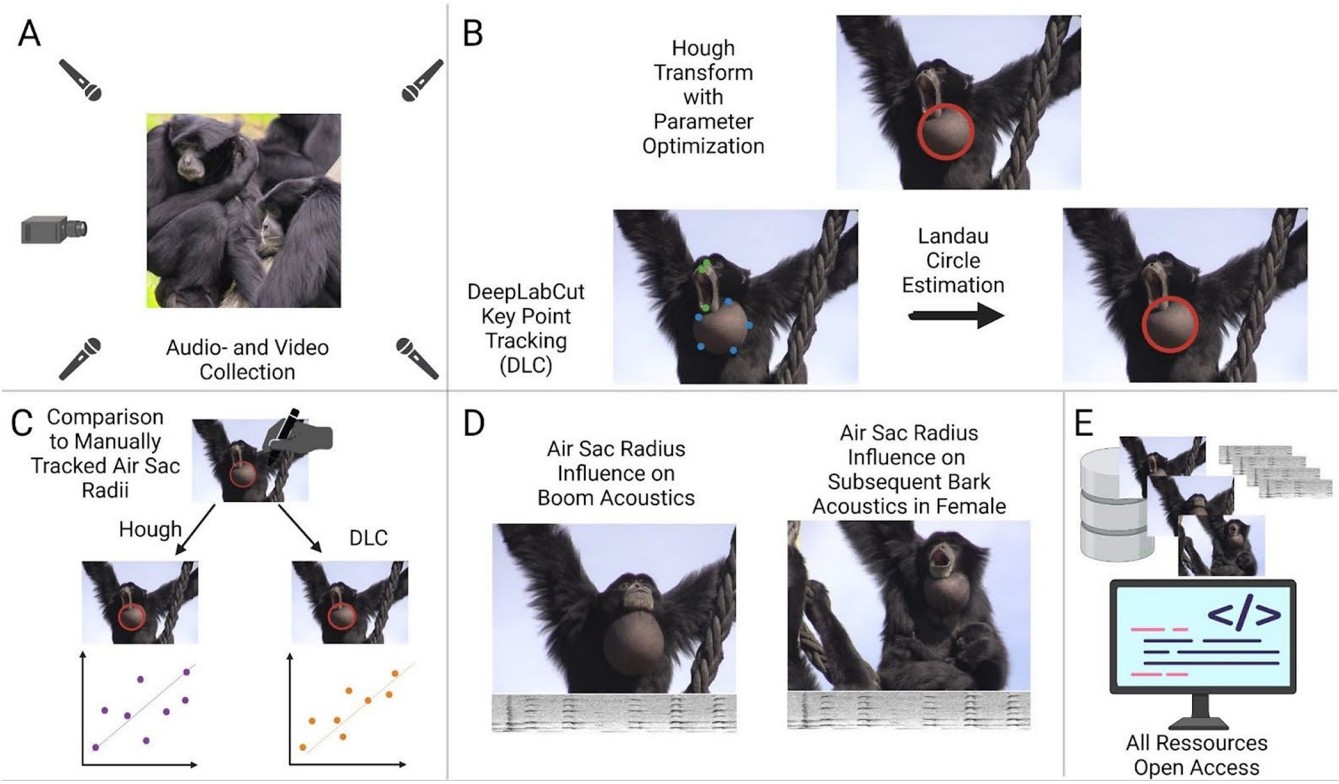

**Fig 2. Overview experimental design.** A) Audio and video data were collected in the Jaderpark Tier- und Freizeitpark an der Nordsee, Germany. B) Air sacs were automatically tracked with two approaches: the lesser performing Hough Transformation (see sample here) versus the very good performing DeepLabCut tracking (see here sample) with Landau Circle Estimation (DLC+: see here for a sample). C) For a subset of the data, air sacs were tracked manually and compared to the automatically tracked radii. DLC estimated radii had a high correlation of r > 0.8 with the manually tracked radii. D) Acoustic parameters of two different kinds of calls were analyzed and related with air sac inflation as a proof of concept. E) All data and code are shared open access. The figure was created with BioRender. All images are photos by Wim Pouw and Mounia Kehy.

animal testing as the study did not change the circumstances of the captive siamang in any way.

## Overview

**I)** Audiovisual recordings of siamang in captivity were made which forms the open dataset of the toolkit (see Audiovisual Dataset). **II)** We investigate two approaches to track the recorded air sac inflations automatically: 1) circle tracking through mathematical transformations with the Hough Transform, and 2) circle tracking through point tracking using a trained DeepLab-Cut model (version 2) combined with circle estimation using the Landau algorithm [50], which we refer to as DLC+. **III)** To show the toolkit in action, we provide two different proof-of-concept analyses on the corresponding acoustic and newly obtained kinematic data. The experimental design is summarized in Fig 1. Together, our toolkit and proposed analysis pipelines allow for the extensive study of air sac dynamics, with further investigations of articulatory rhythms (the dataset contains labial kinematics).

## I) Audiovisual dataset

Audiovisual recordings of six siamang (Table 1) were collected by us at a single location in Germany, Jaderpark Tier- und Freizeitpark an der Nordsee, for 21 research days over two

**Table 1. Information on individuals: Sex, age class, and age for all siamang at Jaderpark are given.** Data were only used from five of the six individuals; Tristan, the newborn, was not considered.

| Individual | Pelangi | Roger | Baju | Fajar | Jamil | Tristan |
|---|---|---|---|---|---|---|
| Sex | f | m | m | m | m | m |
| Age class | Adult | Adult | Subadult | Juvenile | Infantile | newborn |
| Age | 18y10m | 29y10m | 7y8m | 4y11m | 3y6m | 6w |

months in the summer of 2022. We used an opportunistic sampling scheme. The opportunistic sampling started whenever the apes began to sing. The apes usually sang in the mornings, after lunchtime, and/or occasionally around five p.m. For the sampling of singing events, we randomly picked one of two recording strategies for each recording session: 1) record whoever is best visible, or 2) record an individual chosen randomly. The first strategy maximized the amount of usable close-up video recordings, while the second strategy allowed for tracking the song's development as contributed by a single individual (although it did lead to a lot of unusable video data due to occlusions). The closeup audiovisual recordings can be accessed on the Donders Repository (https://data.donders.ru.nl/collections/di/dcc/DSC_2022.00071_151?3).

## Audiovisual recording

A tripod-operated camera was used (Canon Legria HF G30) with high-zoom capabilities due to an add-on lens (TL-H58 Telekonverter), sampling at 25fps (50i), and for our second visit, we sampled at 50fps to increase temporal resolution. A Sennheiser ME64-12 was fed as an audio input to the camera (using a DXA-2T audio adapter).

## Audio recording

We used two types of microphones. Firstly, a cardioid boom microphone with a windjammer was directed at the center of the site (Sennheiser ME67 HDP2), which was connected to a DR-40 linear PCM recorder (TASCAM) sampling at 48Khz.

Additionally, we recorded multi source audio streams from four KE400 Sennheiser microphones with windjammers sampling at 48 kHz at four locations. We combined all channels into a single mono-channel source by synchronizing audio waveforms using Adobe Premiere Pro CS6. This multi source audio stream is more suitable for estimating acoustic measurements, such as amplitude, as amplitude is influenced by the distance and direction between the microphone and the sound source.

## Enriched data

The data repository features information about the weather (humidity, temperature, cloudiness, etc.) during each recording session, next to the start and end times of the recordings. Additionally, pictures are provided with the names of the individuals. Finally, we have tracked all videos in the repository with our DLC+ model using a GPU-supported machine. These tracked videos and resulting time series data are also in the repository.

## II) Computer vision tracking tools

**Ground truth.** To assess the success of automatic tracking, we first established a ground truth, where we asked a student assistant to manually track the radii of air sacs in images of siamang closeups. We created a subset of 1612 frames from nine scenes and three days (three scenes per day) to account for different lighting and backgrounds. Radii were drawn on individual frames in the OpenSource Software Fiji, and the diameters and coordinates of the circle

center were exported. If no air sac was visible, or deemed to be untrackable because of occlusion, low inflation, or any other reason, we drew a very small circle at the edge of the frame, clearly smaller than any tracked circle. This systematic strategy allowed to include those 'untrackable' frames in testing the automatic tracking. Before comparison, we transformed diameter to radius.

**Unsupervised computer vision: Hough transform.** We use a feature extraction technique called the Hough Transform to detect imperfect circles (or semi-circles) in individual video frames [50]. The complete process is implemented in Python and primarily supported by OpenCV (see script here). In the procedure, frames undergo a series of preprocessing steps that are commonly advised to increase the detection success of circles, which essentially serve to maximize the visibility of any edges in the scene. Our Hough detection and the preprocessing process is depicted in Fig 3, where circles detected with the Hough Transform are indicated for all processing steps to illustrate their necessity. First, frames were converted to grayscale and optimized for brightness and contrast. A median blur is applied as a next step, followed by a 'Canny edge detection', where the found edges are then dilated. The Canny edge detector inputs a lower and an upper threshold in which edges are detected. The issue arose that the threshold is easily set too high or too low for a particular scene. Generic thresholds for very different images increase this problem; ideally, the thresholds are tuned for each video separately. Therefore, we set the lower and upper limits by a normalization procedure, where the threshold depends on the mean intensity of the frame in question. After dilating, the image is blurred again with a median blur.

After image preprocessing the Hough Transform can be applied to the image. In our case the Hough transform is set to detect semi-circles with a minimal radius of five pixels and a

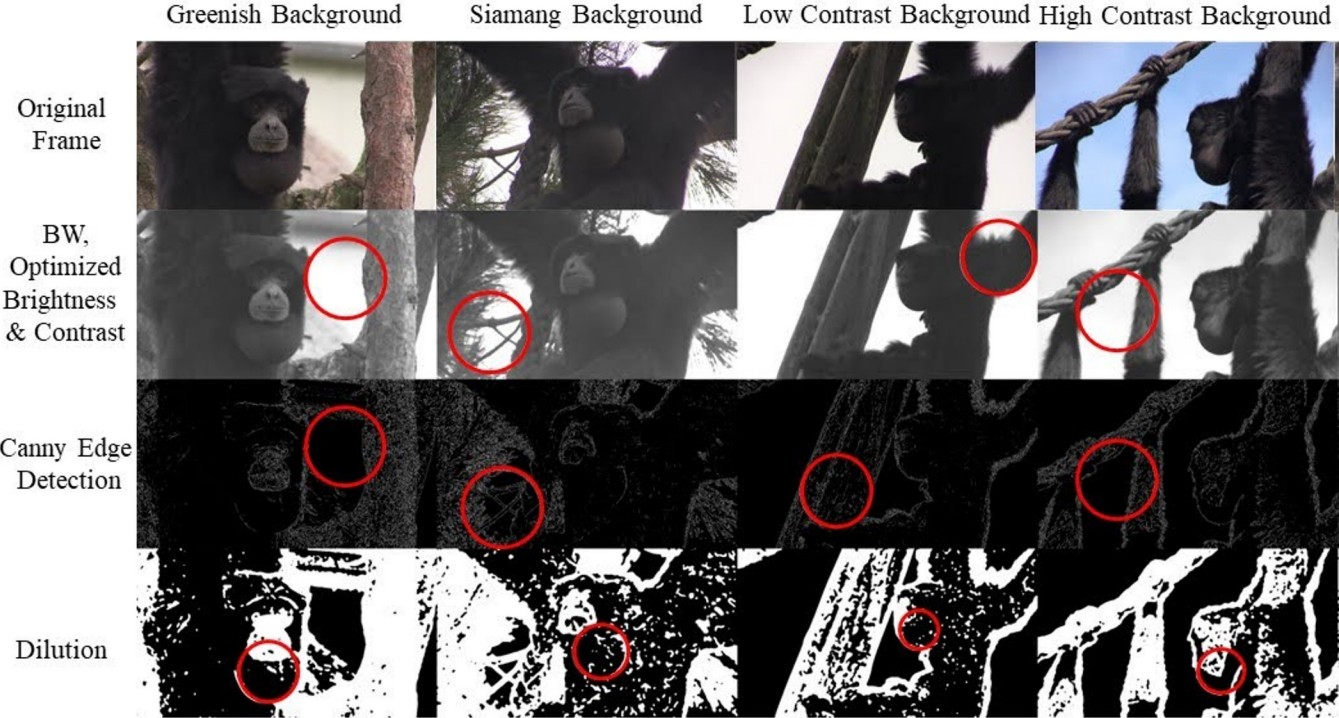

**Fig 3. Image processing to increase Hough Transform circle detection success.** Four pre-processing steps in four different frames with various backgrounds are shown to illustrate the necessity of the preprocessing steps for the Hough Transform approach to work. Only after the last pre-processing step are air sacs correctly found with the Hough Transform. All images are photos by Wim Pouw and Mounia Kehy.

**Table 2. Parameters iterated for parameter optimization in Hough Transform.** After initial testing, six parameters were chosen to iterate over different setting combinations to find the best performance. The best performance was found by correlating the found radius results to the manually tracked radii and choosing the parameter combination, which showed the highest correlation over all videos.

| Parameter | function | Range | steps | Best for our videos |
|---|---|---|---|---|
| Alpha | Brightness | 0.5–3 | 0.5 | 2 |
| Beta | Contrast | 20–40 | 5 | 30 |
| Blur | Median Blur | 25,27,29,35 | | 27 |
| Dilation | Dilation | 3–7 | 1 | 5 |
| Canny 1 | Parameter used to determine lower limit in Canny edge detection | 4–12 | 2 | 5 |
| Canny 2 | Parameter used to determine upper limit in Canny edge detection | 8,10,13,15,17 | | 14 |

maximum of 270 pixels can be found (these numbers depend on your data and aim). We found that 270 pixels as a maximum for detection was optimal for our dataset by trial and error and visual inspection. But for future users this needs to be adjusted on a case-by-case basis, depending on the expected circle size in the videos. To reduce any noise-related jitter, after per-frame circle detections in the full video, we applied a Kolmogorov-Zurbenko smoothing algorithm to the detected position x,y data (centroids) and radii of circles.

The preprocessing parameters were optimized with another Python script, changing brightness and contrast parameters, median blur, edge detection parameters, and dilation strength (Table 2). This optimization is necessary for each new dataset. Depending on the variability in the dataset, some parameter combinations might perform well for some and poorly for other images; instead, other parameter combinations might deliver a satisfactory performance for all images. Another step for potential optimization of circle detection with the Hough Transform is the correct choice of the image section. Hand-selecting region of interest can increase tracking success. Thus, we advise cropping the video so that the circular structure is maximized in the frame. Fig 4 provides an example of the output of the Hough tracker.

## Supervised computer vision: DeepLabCut + Landau (DLC+)

**DLC model info.** We trained a Resnet-101 convolutional neural network at 500K iterations using Deeplabcut 2.0 [3,4]. We first trained a shallower network that is set as the default in DLC, Resnet-50, which yielded poor air sac tracking and was soon abandoned. The key

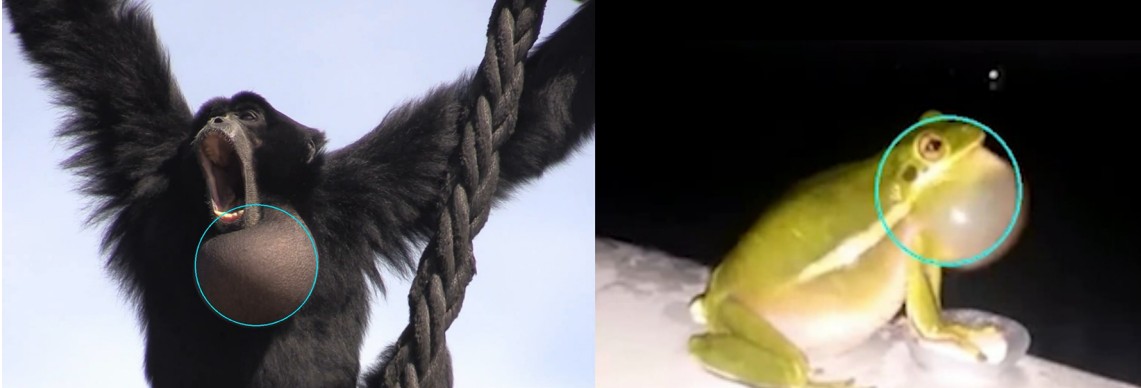

**Fig 4. Example of our Hough Transform approach.** Examples: Hough Transform tool applied to a video of a siamang (see here) and a green tree frog (*Hyla cinerea*; see sample here), with satisfactory performance for some purposes such as position tracking or large-scale changes in inflation (but after investigation of tracking performances we favor DLC+). The left image contains a photo by Wim Pouw and Mounia Kehy. The picture on the right is from the youtube video and analyzed by us for testing.

points that our model tracks are: ['UpperLip', 'LowerLip', 'Nose', 'EyeBridge', 'Start_outli-ne_outer_left', 'Start_outline_outer_right', 'LowestPoint_outline, 'MidLowleft_outline', 'Mid-Lowright_outline']. The model weights and metadata (e.g., training loss data) can be found on our github.

**DLC labeling.** A training dataset was created with a labeling approach that is optimized for 2D tracking of semi-circular 3D objects under variable camera angles. We found the fol-lowing DLC labeling approach provides a robust method for tracking circular objects with DLC+. Firstly, we define the start of the outline on the left of the semi-circle, then the right end of the outline (Fig 5). These are fixed points and can be isolated based solely on visual fea-tures in the frame as they do not require the context of other points to be determined. The other non-fixed points lie on the edge of the air sac but are further defined in relation to the fixed points. The lowest point is defined as a point on the edge of the air sac, on the middle of a line perpendicular to the line between the start left and the end right. Then, the middle points along the circle are defined for the left and right sides. Note, that this labeling method allows for a well-formed description of each point, regardless of the perspective of the siamang (see Fig 5). Only three points are necessary to estimate a circle in the subsequent Landau estima-tion, described below. By aiming to track five points while only three are necessary, the track-ing becomes more error-robust, as the likelihood of two points dropping out of the frame is much lower than a single one dropping out.

**DLC training set and internal DLC validation.** The total number of hand-annotated images was 390, with only one animal prominently in the frame. This set of images was used as a training (90%) and test (10%) set for training of the DLC model. After training (500k itera-tions), we yielded a pixel error for 0.6 likelihood tracked points of 9.5 pixels for the test image set (for 1920*1080 = 2.073.600 pixel images).

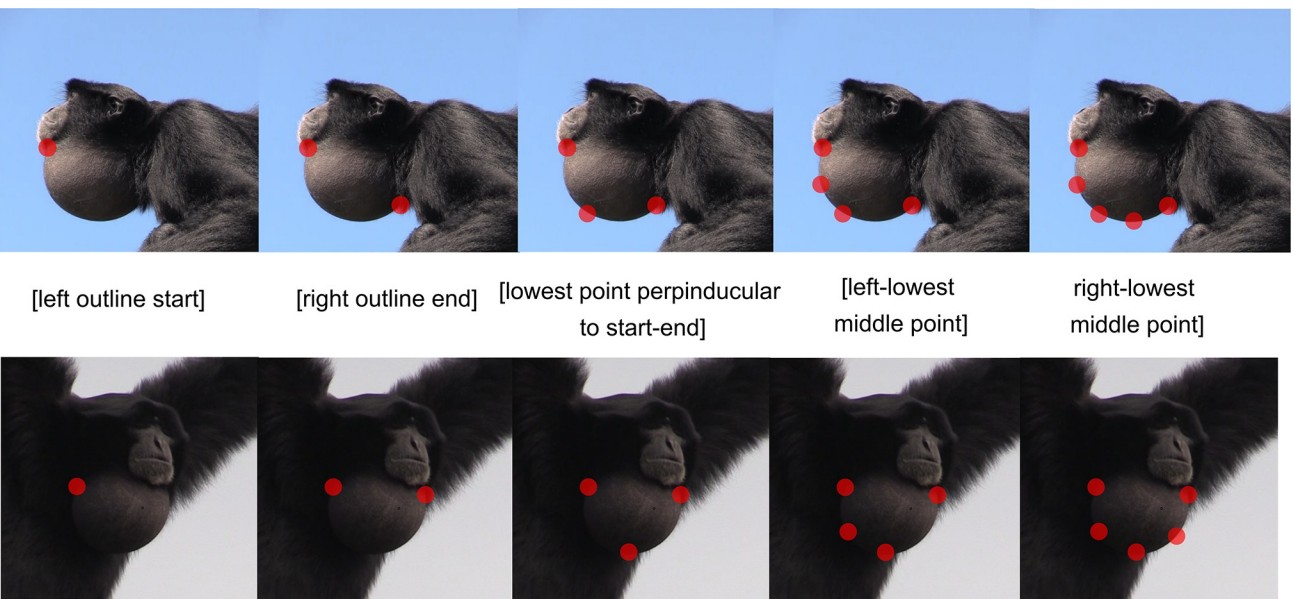

[left outline start]  [right outline end]  [lowest point perpinducular to start-end]  [left-lowest middle point]  right-lowest middle point]

**Fig 5. DLC labeling approach for DLC+.** The five points to be tracked on the air sac outline are depicted. Note that two outline ends, which we call fixed points, are clearly defined. The other three points are defined relative to these fixed points. Firstly, it is the middle point between the start and end point (vertically defined by being on the edge of the air sac). The two other points are again the middle of the start or end and the adjacent middle point. In this way, we ensure that DLC can always determine the points, even when some are not determinable, without relating them to the two fixed start and end points. As such, some points are only relationally but nevertheless systematically definable. Photos taken by Wim Pouw and Mounia Kehy.

**DLC+: Radius estimate using Landau.** We used Landau's geometric circle fit method, which is an ordinary least square estimation method to estimate circles from at least three points. In contrast to other least square estimation methods, it is circle specific, has been described as very robust and a benchmark circle-detection method [51], and it is implemented in proprietary software such as MATLAB [52]. The Landau method minimizes the mean square distance from a fitting curve to the data points, to find the best circle in a fixed-point iterative scheme. The geometric Euclidean distance from data points to the fitting curve was used [50]. We estimated the radius of a circle and the (x,y) coordinates of its centroid. In our analysis points tracked by DLC were only used if they have a likelihood value $> 0.6$. For frames, where less than 3 points fulfill that criterion, we did not get a radius estimate and the frame was excluded from further analysis. DLC tracking was transformed into radii and (x, y)-centroid data with a custom made Python script (link). We also implemented the same routine in an R-script (link).

**Kinematic-acoustic analysis approach.** Data snippets for the kinematic-acoustic analysis were sampled randomly from the full dataset. To be included in the analysis and get reliable radii estimations to be matched to acoustic parameters, snippets for both analyses had to fulfill several criteria: 1) camera angle and zoom were not allowed to change within the snippet; 2) a snippet could contain only a single siamang vocalizing; 3) background noise had to be minimal. Individual vocal boom snippets data were sampled from all four vocalizing siamangs. As great call boom-bark sequences are only or primarily produced by females, samples for this analysis were taken from the only female, Pelangi. Booms produced by Pelangi in the analyzed great call sequences were not analyzed as individual booms but analyzed as part of a sequence.

All acoustic analyses were run in R 4.2.3 [53]. The window length for the acoustic analysis was chosen to match the duration of one video frame. This enables direct comparison of air sac inflation status to the resulting circle-specificities. Video frame data consistently had 25fps. Videos that were originally recorded with 50fps were downsampled to 25fps. This was done after the videos were tracked with DLC. Downsampling, therefore, was performed on the level of analyzed frames. Every second frame with accompanying radius information was kept for the acoustic analysis.

The function "analyze" from the soundgen R-package [54,55] was used to perform an extensive analysis, reporting on 47 standard acoustic parameters, such as amplitude, fundamental frequency (called f0 in the analysis), entropy, or spectral Centroids (see Documentation in soundgen package and codes for a comprehensive list of parameters).

Acoustic analyses were then conducted on two types of calls: a) boom calls and b) bark calls. Acoustic features of the boom calls were compared to the corresponding radius in the same frame, matched by filename and frame number. Pearson's correlation coefficient (r) was calculated for all acoustic parameters and the radius of the air sac with the base R "cor" function. To analyze the influence of the air sac inflation of a boom call on the subsequent bark in a great call sequence (boom–bark—boom—bark—[…]) we extracted the last trackable radius during the boom. We compared it to the average, minimum and maximum of the acoustic parameters across the subsequent bark. We used the same window length for the individual boom call analysis. Acoustics were not analyzed for the boom parts of the great call sequences.

## Results

First, we report the results of the external validation of the air sac tracking methods against ground-truth hand-estimated air sac radii. Secondly, we report the results of the proof-of-concept analyses where we relate acoustic parameters to air sac inflation (referred to as part III of our contribution in this paper).

**Table 3. Reliability comparison with manually tracked ground-truth data between tracking algorithms.** For comparison, only automatically tracked radii below 270 px were used, as this is the maximum radius tracked with the Hough Transform and approximately the maximum radius found manually (max: 266 px). This also applies to smoothed radii. While smoothing increases tracking success for Hough Transform, tracking works equally well, if not better, without smoothing using DLC+ tracking. Notice that these correlations feature the whole subset of data. The Hough Transform works very well for particular examples; DLC+ works better on average and, across the board, has a high performance.

| Circle estimation method | Best Mean Correlation Coefficient (r) | sd** | min** | max** |
|---|---|---|---|---|
| Hough Transform, (best settings overall) | 0.19 | 0.22 | -0.18 | 0.57 |
| Hough Transform, overall smoothed* | 0.23 | 0.33 | -0.31 | **0.8** |
| DLC+, LAN | **0.86** | – | – | – |
| DLC+, LAN smoothed* | **0.85** | – | – | – |

\* Kolmogorov-Zurbenko, iterations = 2, window length = 3 |

\*\* between videos

### External validation automatic circle tracking

To evaluate the tracking success of both approaches (Hough Transform & DLC+), a subset of the data was manually labeled (serving as ground truth) to compare to automatic tracking. We then correlated the automatically tracked radii. The results of both methods, raw and smoothed, are reported in Table 3. The smoothed results are visualized in Fig 6. Automatic trackings of DLC+ are of sufficient quality, showing a correlation coefficient r > .80. DLC + showed the highest correlation to manually tracked data, therefore this method is used for the subsequent analysis in part III.

Note that for the tracked radii with the Hough method we set the maximum radii to 270 px as this was the maximum tracked manually and the maximum trackable radius in the Hough Transform algorithm. Radii below 100 px were not regarded for any of the datasets. Colors denote the different video scenes.

### Analyses relating acoustic parameters to air sac inflation (III)

To study the influence of the air sac inflation on the acoustic parameters of co-occurring vocalizations, 47 acoustic parameters were considered for two different call types. We analyzed acoustic parameters for the "boom" calls, produced during air sac inflation and for "bark" calls produced in so-called "great call" sequences by the female siamang. Barks are produced directly after a boom in the sequences that we selected for analyses (see an example here; note, barks would be extracted from this longer sequence). We matched acoustics to video data by frame, to compare air sac inflation and acoustic parameters in a meaningful way.

### Air sac inflation influences acoustic parameters of boom call as predicted by the model in adults

We correlated air sac inflation as determined with DLC+ for two adult individuals (one female) with a set of acoustic parameters provided by the soundgen R-package [54,55]. A total of 25 call sequences were analyzed with 176 adequate frames (M[SD] frames per call sequence = 7[5]). This final set was the number of frames for which we could determine an air sac radius and concurrent acoustic parameters of the boom call.

Several of the tested acoustic parameters showed a significant correlation with the radius of the air sac (Fig 7E, correlation plot adults). Fig 7, top panel, left shows the four parameters amplitude, Fundamental Frequency (f0), entropy and spectral centroid, in more detail. Amplitude indicates the sound pressure level of the signal while f0 as we use it here stands for the fundamental frequency representing the lowest frequency component of a complex sound

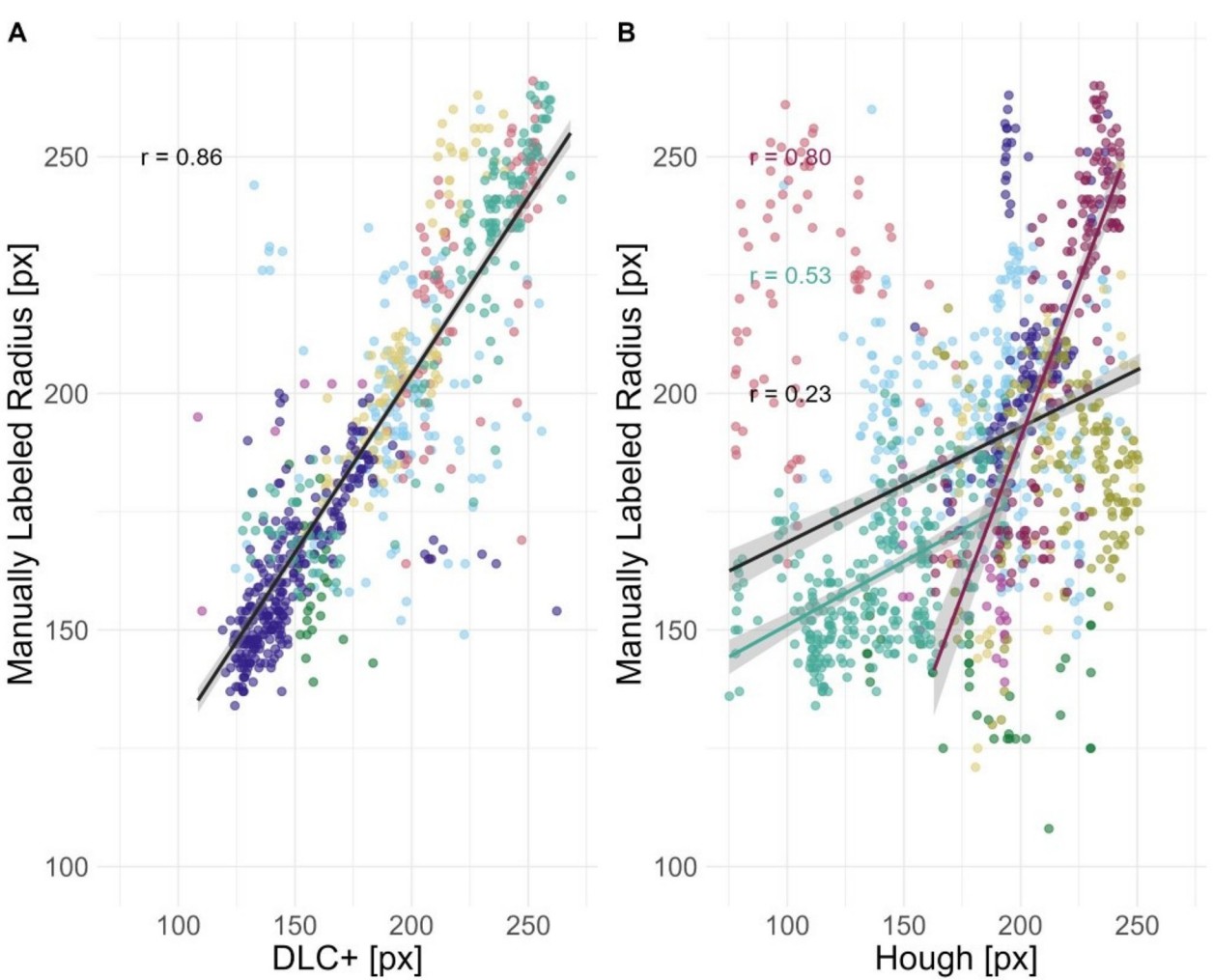

**Fig 6. Automatically tracked radii in comparison to manually labeled radii for both approaches: DLC+ and Hough Transform.** A) Comparison of DLC+ trackings and manual tracking of air sac radii, both in pixel units, for a set of > 1000 frames from nine different video scenes. Perfect fit would mean r = 1.0 and direct match in pixels. We find that the DLC+ radii match very well with the manual trackings, *r* = 0.86, as calculated on all frames over nine videos. B) Comparison of Hough Transform tracking and manual tracking for the same set of > 1000 frames. The correlation coefficient (r) for the nine test videos was 0.23. Parameters can be optimized for each video specifically, and when set adequately for individual videos, we see correlations close to the one for DLC+ trackings. As a trendline, in turquoise, we see the second-best correlation for one video with *r* = 0.53; in red, the best correlation with r = 0.8.

signal. The spectral centroid is a weighted average or the "center of mass" of all frequencies in the signal, containing information about timbre and f0. Finally, (Shannon) entropy is a measure of the disorder in a signal. In bioacoustics, entropy is often used to characterize the complexity or diversity of sound signals. As predicted by theoretical models, we observed an increase in amplitude with an increase in radius (r = 0.45, p < 0.0001***) [34]. The fundamental frequency also strongly correlated with an increasing air sac inflation status (r = 0.82, p < 0.0001***). Entropy and spectral Centroid of the boom call were negatively correlated with radius inflation, though (entropy: r = -0.31, p = 0.002**; spectral Centroid: r = -0.55, p < 0.0001***).

Individual patterns emerge when study subjects are analyzed individually. The overall correlation appears to be driven by the adult female (Fig 7, bottom panel, left). Given our tiny sample size, this individual analysis only suggests that data for more individuals from both sexes is needed to achieve reliable conclusions and test for potential sex differences.

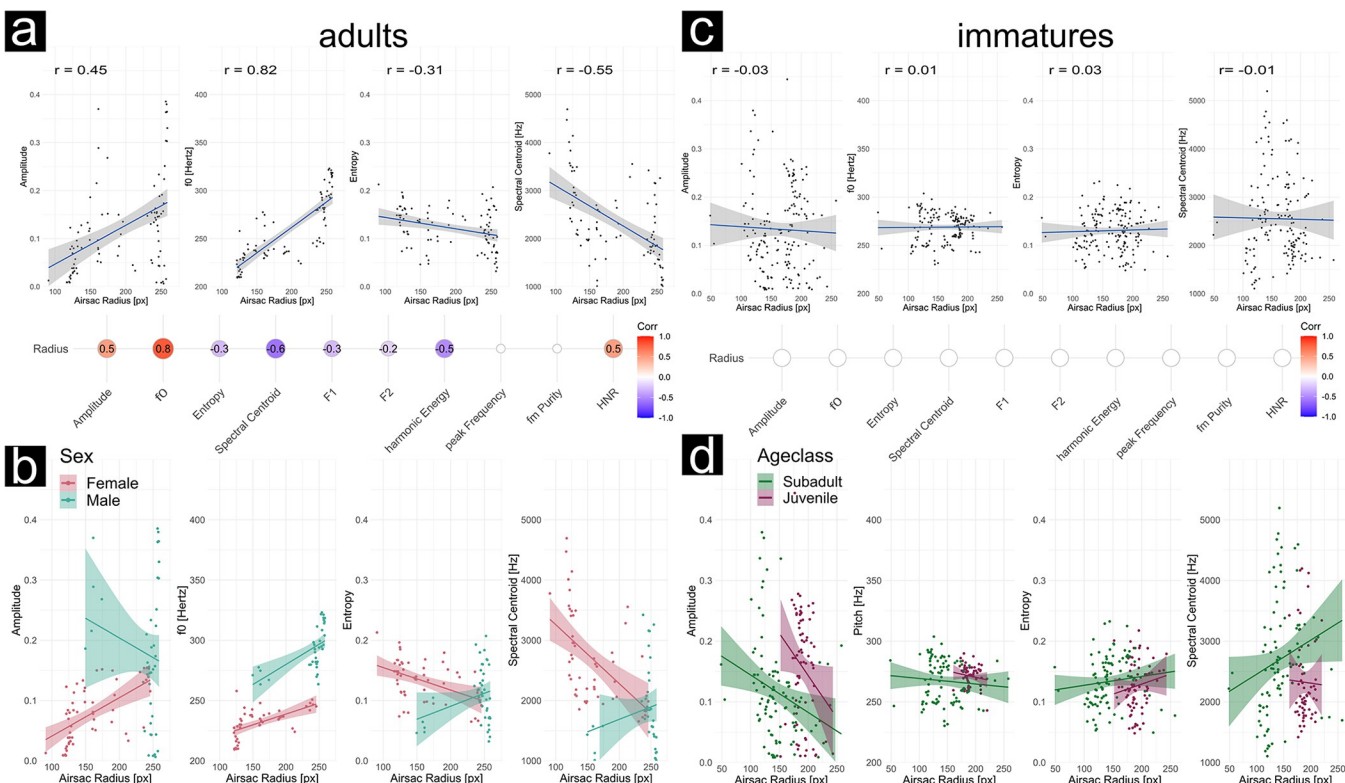

**Fig 7. Significant correlations between air sac inflation (as air sac radius in pixel) and acoustic parameters.** The figure is divided into results of adult siamangs (left; panel a and b) and immature siamangs (right; panel c and d). The top panel shows the pooled data. Air sac radius in pixel on the x-axis is shown against four different acoustic parameters (from left to right): sound amplitude, f0 or fundamental frequency in Hertz, Wiener entropy and spectral Centroid in Hertz. For adults, we find clear correlations: The more the air sac is inflated, the higher the sound amplitude produced. This is in line with model predictions (see [34]). The fundamental frequency (f0) is also positively correlated with the air sac inflation, and seems to be the only stable relationship reflected in both the male and female individual. Mean Entropy is negatively correlated with air sac inflation, meaning, the more inflated the air sac is, the more tonal the produced sound. The Spectral Centroid is negatively correlated as well, indicating the higher the inflation, the more energy in the lower frequencies. We do not see those relationships in immature not fully grown individuals. In the middle panel significant correlation coefficients for all tested acoustic parameters are shown. Notice, that none of the acoustic parameters showed a significant relation in immatures (indicated by white circles). The bottom panel divides the pooled data. For adults we divide by sex into male and female, for the immatures we divide by ageclass into subadult and juvenile.

In addition to the adults, two male individuals of younger age classes were analyzed: Baju, a subadult (7 years, eight months), and Fajar, a juvenile (4 years, 11 months). Their results contrast the results we found in adults and what theory predicts. No clear pattern emerged when we pool both age classes (Fig 7, top panel, right). When dividing further by age class (juvenile and subadult), we can see patterns opposite to what we see in adults and expected to see from adults and modeling work (Fig 7, bottom panel, right).

This difference between the patterns across age classes suggests an influence of ontogeny on the relationship between acoustic parameters and air sac inflation that should be considered in further analyses to see if the effect is stable with a larger sample size and what this could mean.

## Influence of air sac inflation on subsequent calls

In a second analysis, we studied how air sac inflation of a boom call influences the acoustics of the subsequent bark, using early phases of the great call sequences of the only female Pelangi (which consisted of these alternating boom-bark units, see here). We analyzed 16 boom-bark pairs from a total of 8 great call sequences. We found that the air sac inflation statistically predicts the average spectral centroid of the subsequent bark (Fig 8B): The more inflated the air

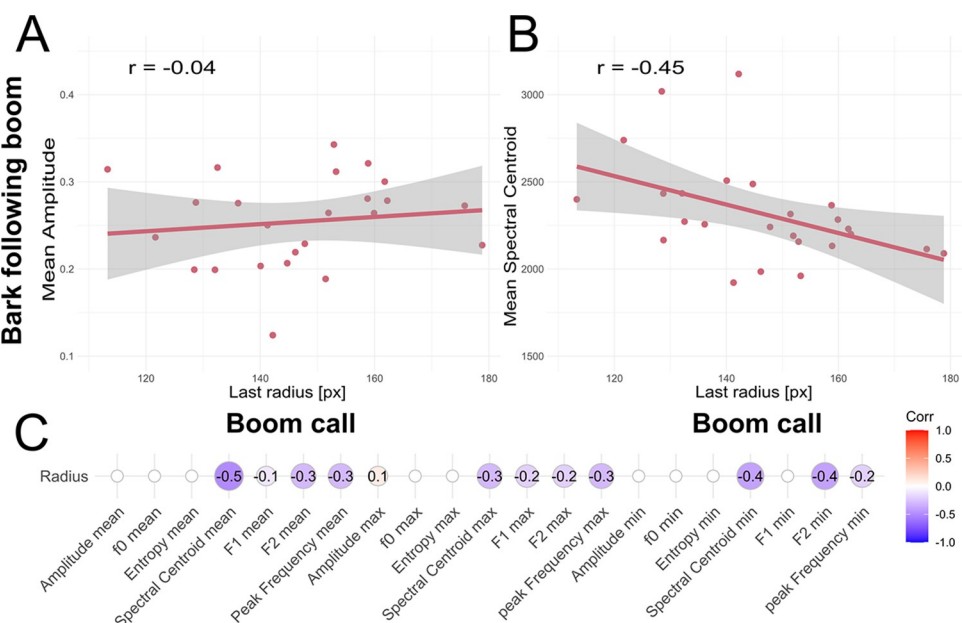

**Fig 8. Relation between air sac inflation for a boom call, with acoustics of bark immediately following the boom.**
The top panel shows the relationship between the last observed radius of the air sac during a boom call, plotted against the subsequent bark's acoustic parameters. A) shows the results for the acoustic parameter mean sound amplitude, which does not relate to the inflation of the air sac in this case. B) shows the relation of the mean spectral centroid (given in Hertz) of the bark relative to the inflation of the preceding boom. In C), correlation estimates are shown for the suite of acoustic parameters included in these exploratory analyses, but only statistically significant correlation coefficients are shown. Mean values and minimum and maximum values of the acoustic parameters in the bark were tested against the radius of the last boom.

sac was before the booming call (last trackable inflation radius of the boom), the lower the average spectral centroid of the bark (r = -0.45, p = 0.001). This matches the relation between these features within one boom call as described above, suggesting that air sac inflation can be a preparatory action for the subsequent call) produced with an open airway. Furthermore, this lowering the spectral centroid aligns with the theory that air sacs attenuate higher formant energy to increase the acoustic appearance of body size and sound radiation [34]. None of the other relations found for boom calls (i.e., entropy and fundamental frequency) could be observed in the data (Fig 8C). No clear correlation to the sound amplitude of the bark call could be found (r = -0.04, p = 0.14, Fig 8A), which speaks against the glottal shock theory suggesting that there is an increased air pressure release due to an extra subglottal pressure producing air reservoir provided by the air sac [33].

## Discussion

The current report breaks terrain in tracking elastic kinematics in animals; it uses unsupervised and supervised computer vision methods to detect semi-circular soft-tissue structures. In our main use case, we focused on the laryngeal air sacs in the siamang (*Symphalangus syndactylus*). We thereby also provide a data archive of 7+ hours of siamang singing with closeup video data with multi-source audio, ideal for the study of articulatory, acoustic, and air sac states. Data sharing in primatology is rare, and we are confident that this openly accessible dataset enriched with DLC+ trackings will be of significant utility to forthcoming researchers due to its user-friendly nature. DLC+ was the best method as assessed with ground truth data. However, our toolkit provides both unsupervised (Hough Transform) and supervised

computer vision tools (DLC+) to track elastic circular biological structures. Finally, summarized below, we provided a kinematic-acoustic analysis of air sac inflation and its relation to the acoustics of calls. Together, this report breaks new terrain that we hope will invite further research incorporating a complete morphometric analysis of animal behavior involving bones, elastic and expandable soft tissue.

Our kinematic-acoustic analyses of air sac dynamics in siamang singing confirmed that the lower frequencies are modulated while the higher frequencies are attenuated, captured by a lower spectral centroid. We find this for adult boom calls and barks following booms in the female boom-bark sequences that occur at the early phases of the so-called "great call". We also observed an increased amplitude of the boom calls for higher inflated air sacs. This finding requires further scrutiny as all gibbons sing loudly, but only siamang have large laryngeal air sacs. Further study is required to understand the many possible factors (e.g., see [27,56]) that shape the gibbon's capacity to sing loudly. These findings obtained from dynamic real-world data align with modeling research where air sacs were initialized at different air sac volumes [34]. We further obtain that adults and younger siamang individuals show different relationships between air sac inflation and the assessed acoustic parameters. This developmental effect may have indicated a role of ontogeny. This is not surprising as a physiological explanation: the cavities of smaller air sacs will have different resonant properties due to morphological differences. Furthermore, we observed other exciting patterns in acoustics and air sac inflation, where, for example, boom f0 increases with air sac inflation. With its small sample size, the current exploratory analysis has only limited explanatory strength but already shows promising paths to further the study of air sacs, especially in the siamang.

The current toolkit also provides a boon for collecting new audiovisual data on elastic circular biological structures in the wild, as there are now tested approaches to study such structures. The pipelines provided in our toolkit are fully reproducible and await large-scale applications in a diverse set of elastic circular biological structures on this planet (Fig 1). The combinations of point estimates using DLC and Landau circle approximation, as implemented in our DLC+, provide the best performance; we recommend them for further research endeavors. We encourage researchers to optimize further tools like the Hough Transform implementation, which we found suboptimal for our current aim. With some adjustments in the preprocessing of images, future studies can improve the Hough Transform approach to the point where it becomes more efficient for large-scale use. In such a case, the need for training a neural network, such as in DLC+, becomes obsolete, sparing labor. Our work incites one more development: training a general-purpose model based on a range of elastic circular biological structures. This trained model could then serve as an out-of-the-box approach for tracking these circular objects, similar to models already developed (see DLC SuperAnimal, [57]). A drawback of the current approach is that we have focused on single-animal tracking, and it invites further development of applications for multi-animal tracking; this is relatively straightforward as a multi-animal DLC approach already exists [58].

Researchers can build on the current DLC+ materials in a number of ways. Most pertinently, the current trained model can be applied to a much bigger dataset containing many more siamang individuals; data are relatively easy to collect, as captive siamang are not known to sing less than siamang in the wild. We hope our contribution of an open dataset will eventually lead to the gradual construction of a bigger repository of audiovisual recordings of siamang singing. Furthermore, the current model trained on siamang air sacs can be retrained to be adapted for circular kinematics in other animals. On the technical side, the current pipeline can be scaled up to 3D tracking using triangulation methods whereby multiple calibrated cameras generate 2D videos that can be tracked with the current DLC+ model, which then form

the input for a triangulation pipeline [59] which obtains 3D positions which can then be used to approximate air sac volume with a higher precision.

Our approach, advancing morphometric studies from bony structures to elastic structures, can potentially address diverse questions across various species, encompassing birds, primates, pinnipeds, and frogs. Consider for example that the role of vocal sacs in anuran vocalizations is well documented, but additional functions have been proposed related to respiration, buoyancy control, chemical signaling, or even thermoregulation [19]. In sum, the ability to study vocal sac dynamics in detail will help shed light on this multi-functionality [19]. Furthermore, tracking of semi-circular biological objects could also be used to assess the health and development of the species through pure observation, serving as an indicator of ecosystem health.

To conclude, the current open-source dataset, open-source computer vision tools, benchmarking, and proof-of-concept analysis provide a way to study diverse biological structures (see Fig 1). The current highlighted case of siamang air sacs can help us understand the adaptive functions of these extreme biological modifications. We invite the community to study the dynamic modulation of elastic circular structures in animals.

## Author Contributions

**Conceptualization:** Wim Pouw.

**Funding acquisition:** Andrea Ravignani, Wim Pouw.

**Investigation:** Lara S. Burchardt, Mounia Kehy, Wim Pouw.

**Methodology:** Lara S. Burchardt, Wim Pouw.

**Resources:** Andrea Ravignani, Wim Pouw.

**Software:** Lara S. Burchardt, Yana van de Sande, Wim Pouw.

**Supervision:** Wim Pouw.

**Validation:** Lara S. Burchardt, Wim Pouw.

**Visualization:** Lara S. Burchardt, Wim Pouw.

**Writing – original draft:** Lara S. Burchardt, Wim Pouw.

**Writing – review & editing:** Lara S. Burchardt, Marco Gamba, Andrea Ravignani, Wim Pouw.

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
