## [Decision Letter · Decision Letter 0]

9 Feb 2024

Dear Mr. Pouw,

Thank you very much for submitting your manuscript "A computer vision and data toolkit for the dynamic study of air sacs in Siamang with a general application to the study of elastic kinematics in other animals" for consideration at PLOS Computational Biology.

As with all papers reviewed by the journal, your manuscript was reviewed by members of the editorial board and by several independent reviewers. In light of the reviews (below this email), we would like to invite the resubmission of a significantly-revised version that takes into account the reviewers' comments.

You will see that two reviewers have now carefully considered your work. While both reviewers commend the paper, they also raise several concerns that require major revisions before the manuscript can be considered for publication. I would kindly request your careful attention and deliberation on these points.

We cannot make any decision about publication until we have seen the revised manuscript and your response to the reviewers' comments. Your revised manuscript is also likely to be sent to reviewers for further evaluation.

Sincerely,

Adriano Lameira

Guest Editor

PLOS Computational Biology

Natalia Komarova

Section Editor

PLOS Computational Biology

Dear Wim Pouw and colleagues,

Thank you for submitting your manuscript to PLoS Computational Biology, we appreciate your patience while we gathered reviews to your paper.

I have now received reports from two reviewers who have carefully considered your work. While both reviewers commend the paper, they also raise several concerns that require major revisions before the manuscript can be considered for publication. I would kindly request your careful consideration of these points.

Reviewer's Responses to Questions

**Comments to the Authors:**

Reviewer #1: I have read with great interest the submission by Burchardt and colleagues.

I applaud the authors for making the data available. I’ve looked at the dataset myself and look forward to hopefully discussing some future possibilities with the authors.

I am not an expert on the methods used, and do not have many comments where the empirical work is concerned. I am, however, well read on the topic of primate air sacs, as well as relevant paleoanthropological work, and note a number of inconsistencies in the literature review and summaries. I hope the authors find my comments useful toward more appropriately framing their work. Overall, my notes are relatively minor.

L99-102

Here, the authors motivative their work by noting the loudness of siamang songs. However, other Hylobates also sing loud songs, and possess no throat sacs. Indeed, this a core argument of Harrison’s (1995) as why air sacs are relatively “functionless” in extant great apes (or, perhaps more appropriately, that they do not have any obvious connection to vocalizations) (this is discussed in my comment on L112). But isn’t this problematic for the authors’ analysis? I think the phrasing in this section is somewhat obfuscating here. In the name of fairness, should not identical studies be conducted to investigate any differences between (non-siamang) gibbon and siamang songs? The authors push the angle that few empirical works have attempted to make sense of effects of air sacs; surely, this comparison is one such necessary work?

L106

I'd suggest complimenting this description of how "the air sacs are inflated" with a description of how air actually enters the sacs. Descriptions of likely mechanisms of air sac distension are provided by e.g., Hill and Booth (1957, p. 320) and Lieberman (2011, p. 320).

L109-10

The authors write, “Laryngeal air sacs often get infected and it is not uncommon for an animal to die from that.” Toward this point, the authors cite a case study by Hastings (1991) describing the successful(!) treatment of air sac infection in a gorilla. This reference does not support the claim. In addition, the sentence is also somewhat awkwardly phrased, and should probably be re-written.

There are certainly many descriptions air sac infection in the medical primatology literature – but I’m not aware of any relevant quantification: what makes death from infection “not uncommon”? Part of the motivation of the paper is that “Air sacs evolved even though the risk and cost of having them is high” but that statement is not well supported by the literature review as is. The authors cite Hewitt et al. (2002). Hewitt and colleagues briefly review something to this effect (p. 73), but the main paper explores wholly different aspects of behavior potentially related to the sacs. I would recommend the authors consult Lowenstine and Osborn (2012) – a work that *does* provide a thorough overview of infections of the primate airways and laryngeal complex.

L112

The authors cite Harrison’s (1995) book among the “many hypotheses about the function of (siamang) air sacs”. Harrison’s hypothesis is essentially a “null” hypothesis – sacs are “functionless” (it should be said that in the relevant section H is concerned with apes, not all primates). That’s not to say that I agree with Harrison’s hypothesis, but seeing how the authors progress from this statement to siding immediately with a “size exaggeration hypothesis”, I’d suggest noting specifically that Harrison’s view effectively offers a null hypothesis, by which results can be evaluated.

L124

Missing letter in “nex*t*”

138-39

"... we can start to better account for variations in vocal acoustics across species". The Fitch "speech-ready" paper should not be cited here. It presents an "vowel space" greatly and arbitrarily inflated by the inclusion of yawning (see Everett, 2017), which involves extreme contortions of the mandible never observed in actual vocalization (e.g., Story et al., 2001).

Why not cite a paper that bases its conclusions on properties of vocalizations by species that possess prominent air sacs, and which may accordingly benefit from the findings presented in the authors' paper? There is no shortage of these in the relevant literature, concerning New World monkeys (e.g., Ybarra, 1986; de Cunha et al., 2015), Old World monkeys (Owren et al., 1997) and great apes (Lameira & Wich, 2008; Hedwig et al., 2014). Howler monkey (Ybarra, 1986) and orangutan (Lameira & Wich, 2008; Ekström et al., 2023) calls are particularly strong candidates (in my own opinion, anyway).

L141-42

(1) The Sima de los Huesos H. Heidelbergensis hominins are generally considered early Neanderthals, and should perhaps not be written out as a separate species without this context, as it is misleading.

(2) The binomial name for Neanderthals is H. neanderthalensis, not “Neanderthalis”.

(3) Genus names should be written out: H. neanderthalensis, not Neanderthalensis.

(4) The sentence is awkwardly written, “… why air sacs seem to have been lost in Neanderthalis, Heidelbergensis, and humans…”, and implies sacs were lost independently in these species – which, again, are more likely realistically seen as two species – not three! (If “Heidelbergensis” lost air sacs, Neanderthals would not have needed to: the first is likely an early form of the second.)

I appreciate the authors are bioacousticians and not paleoanthropologists – but these details need to be corrected. All these issues can be easily rectified by changing the sentence to read, “lost in Homo”.

L142

The idea that “Australopithecus” possessed air sacs is popular in “evolution of language” circles, but empirical support is extremely limited. The conclusion is based on a single(!) hyoid bone specimen found at the Dikika site, Ethiopia (Alemseged et al., 2006). The individual is a juvenile (appr. Three years old) likely female. In extant great apes, including humans and chimpanzees, goes through marked changes in shape as the animal matures. But more importantly, the genus name Australopithecus afarensis must be written out, as there is no evidence whatsoever suggesting presence or lack of air sacs in any other australopith (A. africanus, A. garhi, A. sediba, etc). Reflecting the extreme lack of relevant data, specificity is warranted.

L164

Figure 2a shows gibbons, not siamangs. (Yes, I’m aware BioRender does not currently offer illustrations of siamangs!)

L287

I believe definitions of entropy and spectral Centroid are in order here - in particular, what does it mean for the presented work that (L291-92) "Entropy and spectral Centroid of the boom call are negatively correlated with radius inflation". I can understand this section fairly straightforwardly - however, it bears considering that the audience for a paper heavily centered around the vocal behavioral displays of a primate may struggle.

L300-303

I am not following the argument as to how the “patterns opposite of what was expected” (I’d appreciate if this was spelled out, “such that …”) “suggests an influence of ontogeny…” Does it? The relevant measurements are from two individuals – are these results strong enough to suggest anything concrete? I of course understand the need to point out future work, but this section is unclear to me!

L375-76

“…new ground can be broken to understand the role of laryngeal air sacs in primate communication.”

This statement is inappropriately categorical, without being elaborated. There are several types of air sacs (overviews in Ybarra, 1995; Dixson, 2008; Nishimura, 2020) and neither the organs themselves nor any relationship they may (or may not) have to vocalizations and vocal behavior can at this stage be taken for granted. For example, Many primates that possess them - Papio, Gorilla, Pan, etc. - do not possess visible sacs, but nonetheless perform extraordinary vocal behaviors. Would these methods shed any light on these phenomena? If so, how?

There are also a great number of curiosities in the literature. Miller (1941) dissects the air sacs of a silverback gorilla, and notes that they are not bilaterally symmetrical: one side is much smaller than the other. Would the methods shed light on this?

I understand the authors want to push their method and emphasize its usefulness (and they should, it is a worthwhile effort) but this statement should be qualified with reference to the fact that air sac morphology is highly variable between primates. The study title outlines "a general application to the study of elastic kinematics in other animals", but this is seemingly in reference to e.g., gular sacs in birds. The applicability to primates more generally is far from obvious to me; currently, it seems like an exaggeration. I hope the authors will convince me otherwise!

***Final judgement***

I am happy to endorse publication of this manuscript, should the necessary edits and clarifications be made.

References

Alemseged, Z., Spoor, F., Kimbel, W. H., Bobe, R., Geraads, D., Reed, D., & Wynn, J. G. (2006). A juvenile early hominin skeleton from Dikika, Ethiopia. Nature, 443(7109), 296-301.

de Cunha, R. G. T., de Oliveira, D. A. G., Holzmann, I., & Kitchen, D. M. (2015). Production of loud and quiet calls in howler monkeys. In M. M. Kowalewski, P. A. Garber, L. Cortés-Ortiz, B. Urbani, & D. Youlaos (Eds.), Howler monkeys: adaptive radiation, systematics, and morphology, 337-368.

Dixson, A. F. (2008). Primate sexuality: comparative studies of the prosimians, monkeys, apes, and human beings (2nd Ed). Oxford University Press, USA.

Ekström, A. G., Moran, S., Sundberg, J., Lameira, A. R. (2023). Phonetic approaches to analyses of great ape quasi-vowels, in R. Skarnitzl & J. Volín (Eds.), Proceedings of the 20th International Congress of Phonetic Sciences (pp. 3076-3080). Guarant International.

Everett, C. (2017). Yawning at the dawn of speech: A closer look at monkey formant space. Retrieved from: http://www.calebeverett.org/uploads/4/2/6/5/4265482/commentary_on_fitch_et_al..pdf

Hastings, B. E. (1991). The veterinary management of a laryngeal air sac infection in a free‐ranging mountain gorilla. Journal of medical primatology, 20(7), 361-364.

Harrison, D. F. N. (1995). The anatomy and physiology of the mammalian larynx. Cambridge University Press.

Hedwig, D., Hammerschmidt, K., Mundry, R., Robbins, M. M., & Boesch, C. (2014). Acoustic structure and variation in mountain and western gorilla close calls: a syntactic approach. Behaviour, 151(8), 1091-1120.

Hewitt, G., MacLarnon, A., & Jones, K. E. (2002). The functions of laryngeal air sacs in primates: A new hypothesis. Folia Primatologica, 73(2-3), 70-94.

Hill, W. C. O., & Booth, A. H. (1957). Voice and larynx in Afridcan and Asiatic colobidae. The Journal of the Bombay Natural History Society, 54, 309-321.

Lameira, A. R., & Wich, S. A. (2008). Orangutan long call degradation and individuality over distance: a playback approach. International Journal of Primatology, 29, 615-625.

Lieberman, D. (2011). Evolution of the human head. Belknap Harvard Press.

Lowenstine, L. J., & Osborn, K. G. (2012). Respiratory system diseases of nonhuman primates. In C. R. Abee, K. Mansfield, S. Tardig, & T. Morris (Eds.), Nonhuman primates in biomedical research, 41-81.

Miller, R. A. (1941). The laryngeal sacs of an infant and an adult gorilla. American Journal of Anatomy, 69(1), 1-17.

Nishimura, T. (2020). Primate vocal anatomy and phsyiology: similarities and differences between humans and nonhuman primates. In N. Masataka (Ed.), The origins of language revisited: Differentiation from music and the emergence of neurodiversity and autism, 25-54. Springer.

Owren, M. J., Seyfarth, R. M., & Cheney, D. L. (1997). The acoustic features of vowel-like grunt calls in chacma baboons (Papio cyncephalus ursinus): Implications for production processes and functions. The Journal of the Acoustical Society of America, 101(5), 2951-2963.

Story, B. H., Titze, I. R., & Hoffman, E. A. (2001). The relationship of vocal tract shape

---

## [Decision Letter · Decision Letter 1]

18 May 2024

Dear Dr. Pouw,

Thank you very much for submitting your manuscript "A toolkit for the dynamic study of air sacs in siamang and other elastic circular structures" for consideration at PLOS Computational Biology. As with all papers reviewed by the journal, your manuscript was reviewed by members of the editorial board and by several independent reviewers. The reviewers appreciated the attention to an important topic. Based on the reviews, we are likely to accept this manuscript for publication, providing that you modify the manuscript according to the review recommendations.

As you will note, the reviewers were very satisfied with the new version, which reads much better. There are still a few minor issues raised by Reviewer 2, which we encourage you to address.  

Sincerely,

Adriano Lameira

Guest Editor

PLOS Computational Biology

Natalia Komarova

Section Editor

PLOS Computational Biology

Reviewer's Responses to Questions

**Comments to the Authors:**

Reviewer #1: The authors have addressed all my comments satisfactorily. I'm happy to recommend that paper for publication.

Reviewer #2: Review uploaded as an attachment

**Have the authors made all data and (if applicable) computational code underlying the findings in their manuscript fully available?**

Reviewer #1: Yes

Reviewer #2: Yes

PLOS authors have the option to publish the peer review history of their article (what does this mean?). If published, this will include your full peer review and any attached files.

Reviewer #1: **Yes: **Axel G. Ekström

Reviewer #2: No

Figure Files:

Data Requirements:

Reproducibility:

References:

---

## [Editor Report · Decision Letter 2]

3 Jun 2024

Dear Dr. Pouw,

We are pleased to inform you that your manuscript 'A toolkit for the dynamic study of air sacs in siamang and other elastic circular structures' has been provisionally accepted for publication in PLOS Computational Biology.

Best regards,

Adriano Lameira

Guest Editor

PLOS Computational Biology

Natalia Komarova

Section Editor

PLOS Computational Biology

---

## [Editor Report · Acceptance letter]

18 Jun 2024

PCOMPBIOL-D-23-01656R2 

A toolkit for the dynamic study of air sacs in siamang and other elastic circular structures

Dear Dr Pouw,

I am pleased to inform you that your manuscript has been formally accepted for publication in PLOS Computational Biology. Your manuscript is now with our production department and you will be notified of the publication date in due course.

With kind regards,

Livia Horvath
